# IRAK2 Downregulation in Triple-Negative Breast Cancer Cells Decreases Cellular Growth In Vitro and Delays Tumour Progression in Murine Models

**DOI:** 10.3390/ijms24032520

**Published:** 2023-01-28

**Authors:** Francesca Ferraro, Anja Steinle, Harini Narasimhan, Andreas Bleilevens, Paula-Marie Stolzenberg, Till Braunschweig, Elmar Stickeler, Jochen Maurer

**Affiliations:** 1Department of Obstetrics and Gynecology, University Hospital Aachen (UKA), D-52074 Aachen, Germany; 2Pathology Institute, University Hospital Aachen (UKA), D-52074 Aachen, Germany

**Keywords:** IRAK2, NF-κB, ERK, triple-negative breast cancer, breast cancer stem cells, endoplasmic reticulum stress, autophagy, apoptosis

## Abstract

Breast cancer stem cells (BCSCs) are responsible for tumour recurrence and therapy resistance. We have established primary BCSC cultures from human tumours of triple-negative breast cancer (TNBC), a subgroup of breast cancer likely driven by BCSCs. Primary BCSCs produce xenografts that phenocopy the tumours of origin, making them an ideal model for studying breast cancer treatment options. In the TNBC cell line MDA-MB-468, we previously screened kinases whose depletion elicited a differentiation response, among which IRAK2 was identified. Because primary BCSCs are enriched in IRAK2, we wondered whether IRAK2 downregulation might affect cellular growth. IRAK2 was downregulated in primary BCSCs and MDA-MB-468 by lentiviral delivery of shRNA, causing a decrease in cellular proliferation and sphere-forming capacity. When orthotopically transplanted into immunocompromised mice, IRAK2 knockdown cells produced smaller xenografts than control cells. At the molecular level, IRAK2 downregulation reduced NF-κB and ERK phosphorylation, IL-6 and cyclin D1 expression, ERN1 signalling and autophagy in a cell line-dependent way. Overall, IRAK2 downregulation decreased cellular aggressive growth and pathways often exploited by cancer cells to endure stress; therefore, IRAK2 may be considered an interesting target to compromise TNBC progression.

## 1. Introduction

In women, breast cancer is the most prevalent and second-deadliest type of cancer. [1]. Depending on its oestrogen, progesterone and human epidermal receptor 2 (HER2) receptor status, it can be categorised into luminal-like A, luminal-like B, HER2-enriched and triple-negative breast cancer (TNBC) [2]. Because TNBC lacks the expression of the receptors commonly targeted in breast cancer therapies, it has limited treatment options [3]. The aggressive nature of TNBC is likely because of the high level of breast cancer stem cells (BCSCs) in the tumour bulk [2]. BCSCs are a subset of cancer cells that can self-renew and differentiate into a variety of breast cancer cell lineages, supporting tumour growth [4]. Our laboratory isolated BCSCs from primary human tumours of TNBC, which were described in recent publications [5,6,7,8]. BCSCs exhibit stem cell potential in vitro, as demonstrated by sphere formation assays and flow cytometry analysis of the CD49f and CD44 expression, and in vivo, because they form xenografts that are strikingly similar to the tumours that they were originally isolated from, both histologically and in terms of their gene expression profile [5,6,7,8].

We have previously performed a kinase screening of the bi-potential MDA-MB-468 TNBC cell line to detect kinases that prevent cancer stem cell differentiation (p. 1, [7]). Four to ten lentiviral short hairpin RNA (shRNA) constructs per target, each of which was tested in three replicates, were used to target 420 kinases; successfully transduced cells were immunocytochemically analysed for myoepithelial keratin 5 (K5) and luminal keratin 8 (K8) expression. This allowed the identification and validation of 11 kinases whose downregulation by at least two differential shRNAs induced a luminal phenotype (K5^−^/K8^+^). Further research revealed that the downregulation of endoplasmic reticulum to nucleus 1 (ERN1) and alpha kinase 1 in MDA-MB-468 inhibited cellular growth and tumour-forming potential (p. 1, [7]). In the kinase screening, interleukin-1 receptor-associated kinase 2 (IRAK2) was identified.

IRAK2 is a pseudokinase that participates in the ‘Myddosome’ formation with other components of the IRAK family upon toll-like/interleukin-1 receptor (TLR/IL-1R) activation [9,10]. The IRAK2 pathway promotes the phosphorylation of mitogen-activated protein kinases (MAPKs) and nuclear factor-light chain enhancer of activated B cells (NF-κB) [10]. A subfamily of MAPK, called extracellular signal-related kinase 1/2 (ERK1/ERK2), induces the expression of genes related to cellular proliferation, such as cyclin D1 [11]. Through its interaction with ERN1, IRAK2 supports the unfolded protein response (UPR), a pathway that encourages cell survival and stress adaptation [12,13]. When ER-stress is present, ERN1 excises the mRNA of the protein known as IRE1-X-box-binding protein 1 (XBP1), which is subsequently translated into a transcription factor that promotes the expression of chaperones and protein degradation factors, thereby enhancing the endoplasmic reticulum’s (ER) capacity for folding proteins [14]. Apoptosis, UPR and autophagy are all tightly regulated pathways in which cells integrate signals and activate UPR and autophagy to block apoptosis if survival is favourable; if survival is unfavourable, apoptosis is no longer inhibited [15].

Because we previously identified IRAK2 in the kinase screening, we wondered if its downregulation could affect the growth of our primary BCSCs and the commercial TNBC cell line MDA-MB-468. We evaluated the impact of IRAK2 downregulation on cell phenotype, proliferation, sphere-forming ability and tumour development. Additionally, the impacted molecular pathways in the context of apoptosis, autophagy and UPR were investigated.

## 2. Results

### 2.1. IRAK2 Is Expressed by Primary TNBC Cells and Its Downregulation Decreases Cellular Growth

We isolated five primary BCSC lines from human tumours of TNBC, which were named BCSC1, BCSC2, BCSC3, BCSC4 and BCSC5 (BCSCs) [5,6,8]. IRAK2 expression was assessed in these primary cell lines and the commercial cell line MDA-MB-468 at mRNA and protein levels (Figure 1A,B), confirming its expression.

Cells were subsequently transduced with lentiviral particles delivering inducible shRNAs-targeting IRAK2 transcripts, and IRAK2 knockdown was induced by supplementing cellular media with 100-ng/mL doxycycline. Although BCSC2 and BCSC5 presented an mRNA knockdown of 31% and 51%, respectively, they did not show phenotypic or proliferative alteration, being excluded from follow-up experiments. IRAK2 was downregulated at the mRNA level of 70%, 65% and 37% in BCSC1, BCSC3 and MDA-MB-468, respectively (Figure 1C), which was confirmed at the protein level (Figure 1D). shIRAK2 cells grew sparser than the respective control cells (Figure 1E), particularly the shIRAK2 BCSC1 cells, which formed tighter and smaller colonies than the respective control cells (Figure 1E: BCSC1). BCSC1, BCSC3 and MDA-MB-468 showed significantly decreased proliferation in the presence of IRAK2 knockdown (Figure 1F). Moreover, to investigate the effects of IRAK2 downregulation on cellular self-renewal capacity in vitro, cellular sphere-forming capacity was assessed, and it was reduced upon IRAK2 downregulation by 33%, 71% and 83% in BCSC1, BCSC3 and MDA-MB-468, respectively (Figure 1G,H).

We previously reported that IRAK2 downregulation induced luminal-like differentiation of the bi-lineage cell line MDA-MB-468, thereby impairing K5 and inducing K8 expression [7]. BCSC1 and BCSC3 present a bi-lineage phenotype as well, expressing both luminal epithelial and myoepithelial keratins [6,8]. Assessing keratin expression in control and shIRAK2 BCSC1 and BCSC3, a significant difference was not observed, suggesting persistent luminal epithelial and myoepithelial characteristics of the cells (Figure A1A,B), suggesting that the effects on proliferation and self-renewal observed in primary BCSCs may not reflect the differentiation phenotype observed in the long-established and homogenous MDA-MB-468 cell line.

### 2.2. In BCSCs and MDA-MB-468, IRAK2 Downregulation Reduces NF-κB and ERK Phosphorylation as Well as IL-6 and Cyclin D1 Expression

NF-κB phosphorylation upon IRAK2 knockdown was evaluated because IRAK2 participates in TLRs/IL-1Rs signalling and the UPR, and both pathways can activate NF-κB [10]. BCSC1 and BCSC3 presented decreased NF-κB phosphorylation following IRAK2 knockdown induction (Figure 2A). NF-κB activation may lead to interleukin-6 (IL-6) expression [16]; therefore, IRAK2 knockdown effects on IL-6 expression were evaluated by performing Western blot analysis. IL-6 was downregulated in shIRAK2 BCSC1 and BCSC3 (Figure 2B,C). Because ERK may be activated downstream of IRAK2 activation [10], we assessed its phosphorylation, reporting that it was significantly decreased in shIRAK2 BCSC3 and MDA-MB-468 compared with control cells (Figure 2D,E). Cyclin D1 expression was assessed because it promotes cell cycle progression from G1 to S phase and IRAK2 downregulation reduced the ability of BCSC1, BCSC3 and MDA-MB-468 to proliferate [17]. Furthermore, cyclin D1 expression can be induced by NF-κB and ERK [18,19]. In shIRAK2 cells, cyclin D1 expression was significantly reduced at the protein level (Figure 2F,G).

### 2.3. ER-Stress Upregulates IRAK2 That Takes Part in the ERN1 Pathway

Previous studies reported that IRAK2 participates in the UPR in the presence of ER stress in prostate cancer cells [12]; therefore, we wondered if IRAK2 could be involved in the ERN1 signalling pathway in BCSCs. ER-stress was induced using thapsigargin in BCSC1 and BCSC3, and the transcription of the UPR-related genes ERN1, XBP1 spliced (XBP1s) and CCAAT-enhancer-binding protein homologous protein (CHOP) was evaluated upon IRAK2 knockdown and thapsigargin exposure (Figure 3A). Cells presented increased transcription of ERN1, XBP1s and CHOP when ER-stress was induced (Figure 3A). Interestingly, ER-stress also increased the transcription of IRAK2 upon IRAK2 downregulation, as well as ERN1, XBP1s and CHOP (Figure 3A). Considering that the transcription of the UPR-related genes decreased in the presence of ER-stress and IRAK2 downregulation in BCSC3, the role of IRAK2 in the activation of the ERN1 pathway was confirmed (Figure 3A). Moreover, ERN1 expression was significantly decreased at the protein level in shIRAK2 BCSC1, BCSC3 and MDA-MB-468 (Figure 3B,C), whereas CHOP expression was affected, although not significantly (Figure 3D,E).

### 2.4. IRAK2 Downregulation Reduces Autophagy in BCSC1 and BCSC3 and Causes BCSC3 to Undergo Apoptosis

Because we noted that IRAK2 is involved in the ERN1 signalling pathway in BCSCs, and the UPR and autophagy are strongly related pathways, we examined IRAK2’s role in autophagy [15]. The quantification of the lipidated microtubule-associated protein 1A/1B-light chain 3 (LC3-II), a common marker used to detect autophagic structures [20], revealed decreased autophagy in shIRAK2 BCSC1 and BCSC3 (Figure 4A,B).

We tested whether IRAK2 knockdown could also affect apoptosis by observing the externalisation of phosphatidylserine (PS), a marker of early apoptosis [21], on the outer side of the cellular membranes, considering that the UPR and autophagy pathways can switch to apoptosis when cells cannot restore homeostasis [15]. BCSC1 and BCSC3 displayed increased apoptosis, with BCSC3 exhibiting statistical significance, following IRAK2-induced downregulation (Figure 4C,D). Moreover, compared with the respective control cells, the expression of the pro-apoptotic protein BCL2 antagonist/killer 1 (BAK1) was decreased in shIRAK2 BCSC1, increased in shIRAK2 BCSC3 and unaffected in shIRAK2 MDA-MB-468 (Figure 4E,F).

### 2.5. IRAK2 Knockdown Delays the Growth of Xenografts Derived from BCSC1 and MDA-MB-468

The tumourigenicity of BCSC1 and MDA-MB-468 in the presence of IRAK2 downregulation was subsequently assessed using a doxycycline feeding paradigm. When BCSC1 and MDA-MB-468 xenografts reached a 3-mm diameter, IRAK2 knockdown was induced by feeding the mice with gelatin containing doxycycline. IRAK2 knockdown was verified at the mRNA and protein levels in BCSC1 and MDA-MB-468 xenografts (Figure 5A,B and Figure A2). BCSC1 and MDA-MB-468 shIRAK2 xenografts were significantly smaller and their growth was significantly delayed compared with the control xenografts (Figure 5C,D). Interestingly, satellite tumours around the primary tumour were a hallmark of MDA-MB-468–derived xenografts, primarily in control animals (Figure 5C, right panels). Four of six control xenografts presented one to three satellite tumours, whereas three of six shIRAK2 MDA-MB-468 xenografts presented only one small satellite tumour (Figure 5C, right panels). Moreover, cyclin D1 expression assessed by immunohistochemistry was reduced in the shIRAK2 BCSC1 xenografts (Figure 5E,F).

## 3. Discussion

We reported that IRAK2 contributes to BCSCs and MDA-MB-468 self-renewal and likely has a pro-oncogenic role. The bi-potential phenotype of BCSCs did, however, continue after IRAK2 was downregulated because the expression of the keratins was unaffected, suggesting that there was no overt differentiation. When we examined the molecular mechanisms underlying the outcomes we observed, we discovered that the presence of IRAK2 knockdown decreased NF-κB and ERK phosphorylation as well as IL-6 and cyclin D1 expression. Considering that NF-κB, ERK and cyclin D1 pathways favour cellular growth [19,22,23], their impairment upon IRAK2 knockdown could explain the decreased proliferation we had observed. Moreover, because cyclin D1 expression was impaired in shIRAK2 BCSC1 xenografts, it may be responsible for delayed tumour growth.

We reported that IRAK2 transcription is upregulated in BCSCs when an ER-stress inducer stimulates them, revealing an IRAK2 requirement in the presence of ER-stress; interestingly, IRAK2 upregulation in the presence of ER-stress agents was previously reported in a primary prostatic carcinoma cell line [12]. IRAK2 knockdown per se affected ERN1 expression and XBP1 splicing, suggesting an interaction between IRAK2 and ERN1 signalling. These data opened up a new perspective related to the role of ERN1 in MDA-MB-468, because the cellular growth inhibition that we reported upon ERN1 depletion (p. 1, [7]), as well as being caused by cellular differentiation, could be addressed in terms of ERN1 implications in the UPR.

BCSC1 and BCSC3 presenting IRAK2 knockdown displayed a significantly decreased autophagy, and considering that ERN1 induces autophagy through the TRAF2/JNK/c-Jun pathway or XBP1 splicing [24], the decreased autophagy could be due to IRAK2-mediated ERN1 downregulation. Moreover, BCSC1 and BCSC3 displayed increased apoptosis upon IRAK2 knockdown, as represented by increased PS externalisation. The downregulation of IL-6 and NF-κB mediated by IRAK2 knockdown may be responsible for the increased PS externalisation, because IL-6, as well as being a pro-inflammatory cytokine, is also an anti-apoptotic factor [16], and NF-κB activation prevents apoptosis in several cell types [23]. Therefore, BCSC3 characterised by IRAK2 knockdown may present increased apoptosis due to IL-6 and NF-κB downregulation by upregulating BAK1, whereas BCSC1 characterised by IRAK2 knockdown may activate apoptosis through different pathways that do not include BAK1 upregulation. These results are consistent with those of previous studies showing that the inhibition of other players in the IRAK2 pathway decreases NF-κB activation, thereby inducing apoptosis [25]. Moreover, IRAK2 downregulation was previously associated with increased apoptosis in canine breast cancer [26]. These findings confirmed that IRAK2 may help cells handle stress-related conditions by stimulating the UPR and autophagy while preventing apoptosis. Therefore, IRAK2 knockdown may impair both the ability of cells to proliferate and self-renew and their ability to resist hostile environments.

The cell lines considered were slightly differently affected by IRAK2 downregulation at the molecular level. Different factors may contribute to the cell line-specific molecular responses detected upon IRAK2 downregulation. MDA-MB-468 is a commercial cell line, whereas BCSCs were recently isolated from primary TNBC tumours [5,6,8]. They present different mutations and genetic backgrounds, and they belong to different TNBC subtypes, given that MDA-MB 468 and BCSC3 are basal-like 1, whereas BCSC1 is basal-like 2. Furthermore, as depicted by our quantitative real-time polymerase chain reaction (qPCR) and Western blot results, the cells present different IRAK2 expression levels and degrees of its knockdown, representative of different residual IRAK2 activities that may differently influence their phenotype.

Cancer cell growth, migration, tumourigenesis and chemoresistance can all be impacted by targeting the proteins involved in the IRAK2 pathway, including IRAK1 and IRAK4 [27,28,29,30]. Targeting UPR-related proteins has also been demonstrated to be a promising anticancer method [12,31,32]. Because we reported that IRAK2 downregulation impaired its pathway and we demonstrated IRAK2 participation in the UPR, IRAK2 could be considered an interesting target to affect the aggressive growth of TNBC.

We concluded that although IRAK2 downregulation affected the molecular pathways in the cell lines considered in this study slightly differently, overall, its downregulation was advantageous because it decreased cellular growth in vitro and delayed tumour progression in vivo. Moreover, IRAK2 downregulation compromised ERN1 signalling and autophagy, which are pathways that are frequently exploited by cancer cells to survive, further supporting the beneficial impact of IRAK2 downregulation in TNBC.

## 4. Materials and Methods

### 4.1. Cell Culture

BCSCs, as previously described [5,6,7,8], were isolated in 2014 from TNBC specimens by the mechanical and enzymatic disruption of the tissue, obtaining single cells that are seeded in 50% matrigel (Corning, 354230, Bedford, MA, USA) at 37 °C and under low-oxygen conditions (3% oxygen, 5% carbon dioxide, and 92% dinitrogen), and subsequently expanded in 2D (2% matrigel) [5,6,8]. BCSCs were authenticated by performing a Multiplex human Cell line Authentication Test. MDA-MB-468 cells (ATCC, RRID:CVCL_0419, Manassas, VA, USA) presented a green fluorescent protein nuclear tag and were cultured in Dulbecco’s modified Eagle’s medium (DMEM, Gibco, 41966-029, Grand Island, NY, USA), 10% foetal bovine serum (without tetracycline to avoid uncontrolled gene downregulation, Gibco, 15140-122, Grand Island, NY, USA) and 1% penicillin/streptomycin (Gibco, 15140-122, Grand Island, NY, USA). All experiments were performed using mycoplasma-free cells. All cell lines were authenticated using single nucleotide polymorphisms profiling within the last 3 years.

### 4.2. Lentiviral Production and Knockdown

One thousand nanograms of the vector of interest were incubated for 30 min in six-well plates (Falcon, Corning, 353046, Durham, NC, USA) with 400 µL of Opti-MEM (Life Technologies, 11058021, Carlsbad, CA), 4 µL of X-tremeGENE™ transfection reagent (Sigma-Aldrich, 6366236001, Taufkirchen, Germany), 700 ng of pCMVdR8.74 (Addgene plasmid, 22036) and 350 ng of pMDVSVG (Addgene plasmid, 8454). Two million 293FT cells (Invitrogen, RRID:CVCL_6911, Thermo Fisher, Waltham, MA, USA) were added to each well, and the medium was replaced with fresh UltraCULTURE™ (Lonza, 12-725F, Basel, Switzerland) 24 h later. The lentivirus-containing medium was harvested and centrifuged at 500× *g* for 10 min 48 and 72 h later. The supernatant was filtered with 0.45-µm filters (Cytiva, WH10462100, Marlborough, MA, USA), overlayed at a 4:1 ratio with 10% sucrose-containing buffer [33] and centrifuged at 10,000× *g* for 4 h at 4 °C. Pellets were resuspended in UltraCULTURE™. The lentivirus was serially diluted 1:2 in DMEM containing 15-μg/mL polybrene (Sigma-Aldrich, 107689, Taufkirchen, Germany) and titrated on 3 × 10^3^ 293FT cells/well of a 96-well culture plate, which was centrifuged at 1000× *g* for 1 h. The selection was performed using 2-μg/mL puromycin (Sigma-Aldrich, P8833-10MG, Taufkirchen, Germany). The highest viral dilution infecting cells was chosen as the multiplicity of infection (MOI) equal to one. Different MOIs were tested on each cell line before proceeding with their infection. Cells were infected with lentivirus at the optimal MOI in a medium containing 15-μg/mL polybrene, and plates were centrifuged at 1000× *g* for 1 h. pLKO-Tet-On vectors were used to perform inducible knockdown [34,35], and the knockdown was induced by supplementing cellular media with 100-ng/mL doxycycline (Sigma-Aldrich, D9891, Taufkirchen, Germany), which was refreshed every 48 h.

### 4.3. Proliferation Assay

Control and shIRAK2 cells were seeded at 3 × 10^3^ cells/well in 96-well culture plates (Falcon, Corning, 353072, Durham, NC, USA), and their growth was observed using the IncuCyte^®^ Live-Cell analysis system (Sartorius, Ann Arbor, MI, USA). After 1-week, growth curves were built considering the ‘cells percentage of confluency’ for BCSCs or ‘green object count per image’ for MDA-MB-468.

### 4.4. Sphere-Forming Capacity

Control and shIRAK2 cells were seeded at 2 × 10^2^ cells/well in 96-well low-attachment flat-bottom plates (Corning, CLS7007-25EA, Durham, NC, USA) in a mixture of 10-µL medium and 10-µL matrigel. Thirty minutes after seeding, 80 µL of the medium was added to each well, and the cells were left to grow for 2 weeks. The doxycycline-containing medium was added to the wells every 2 days. To maintain a constant 100-ng/mL doxycycline concentration, exactly 5 µL of the medium was added to each well every time; however, the amount of doxycycline added was calculated considering the total volume of the medium contained in the well. Sphere-forming capacity was quantified as the ratio between the spheres counted after 2 weeks and the number of cells seeded per well.

### 4.5. Immunofluorescence Staining

Control and shIRAK2 BCSC1 and BCSC3 were seeded at 1 × 10^4^ cells/well in 96-well culture plates. After 5 days, the cells were washed twice with 150-μL/well phosphate buffered saline (PBS), fixed with 100 μL of ice-cold methanol at 4 °C for 15 min, rinsed twice with 150-μL/well PBS (Gibco, 70011044, Grand Island, NY, USA), permeabilised with 150 μL/well of 1× tris buffered saline with tween (TBST) and blocked with 100 μL/well of 1-mg/mL albumin/PBS (blocking solution). The cells were incubated overnight at 4 °C with 40 μL/well of primary antibody diluted in blocking solution; the following day, the cells were rinsed and incubated for 1 h with 50 μL/well of secondary antibody diluted in blocking solution. The secondary antibody was washed out, and nuclei were stained with 4′,6-diamidino-2-phenylindole (DAPI, Sigma-Aldrich, D9542, Taufkirchen, Germany). The following were the primary antibodies used: anti-keratin 5 (Covance PRB, 160P, Princeton, NJ, USA, 1:250), anti-keratin 8 (BioLegend, C5301, San Diego, CA, USA, 1:250), anti-keratin 14 (Covance, PRB-155P, Princeton, NJ, USA, 1:250), anti-keratin 18 (Dako Agilent, M7010, Santa Clara, CA, USA, 1:250) and anti-LC3-II (Cell Signalling, 38689, Danvers, MA, USA, 1:1600). The following were the secondary antibodies used: Alexa Fluor 488 donkey anti-rabbit (Thermo Fisher, A21206, Waltham, MA, USA, 1:500) and Alexa Fluor 568 donkey anti-mouse (Invitrogen, A10037, Thermo Fisher, Waltham, MA, USA, 1:500).

### 4.6. Immunohistochemistry

Pieces of xenografts were fixed in paraformaldehyde (Carl Roth, 0335.1, Karlsruhe, Germany) 4% for 24 h at 4 °C, embedded in paraffin, sliced into 2-μm thick sections and mounted on glass slides. The slides were immersed in xylene for 5 min, then in a descending concentration of ethanol for a few seconds and rinsed with distilled water. The slides were immersed in citrate buffer at a pH of 6 (Dako Agilent, S236984, Santa Clara, CA, USA) for 1 h at 96 °C, allowed to chill for 30 min, washed with distilled water for 5 min and immersed in 5% hydrogen peroxide (Sigma-Aldrich, SA31642, Taufkirchen, Germany)/methanol for 15 min. The slides were incubated overnight at 4 °C with 200 µL of antibody diluent (Dako Agilent, S3022, Santa Clara, CA, USA) containing the primary antibody; the following day, they were washed with Tris-buffered saline 1× for 5 min and incubated for 40 min with 200 µL of the antibody diluent containing the secondary antibody. Next, 200 µL of 3,3′-diaminobenzidine (DAB, Dako Agilent, K3468, Santa Clara, CA, USA) solution was incubated on each slide for 10 min and subsequently washed for 5 min in mineral water and 5 min in distilled water. The slides were subsequently stained by immersing them in hematoxylin (Dako Agilent, C5700, Santa Clara, CA, USA) for 5 min, washing them in mineral water for 5 min and washing them in distilled water for 5 min. The slides were immersed in ascending concentrations of ethanol, bathed in xylene for 5 min and covered with the mounting medium and a coverslip. DAB staining was quantified using the software ImageJ (1.53n, imagej.nih.gov/ij, accessed on 11 January 2023), selecting HDAB in ‘Color Deconvolution’ option. The OD was evaluated using the following equation: OD=Log (maxintensity ÷mean intensity) with max intensity = 255 and mean intensity = mean grey value calculated by ImageJ. The primary antibody anti-cyclin D1 (Invitrogen, PA532373, Thermo Fisher, Waltham, MA, USA, 1:100) was used. The secondary antibody polyclonal goat anti-rabbit (Dako Agilent, P044801, Santa Clara, CA, USA, 1:100) was used.

### 4.7. RNA Isolation and qPCR

RNA was extracted from cell pellets using the miRNeasy^®^ mini kit (Qiagen, 1038703, Hilden, Germany), and reverse transcription was performed using the EvoScript reverse transcriptase kit (Roche, 07912323001, Mannheim, Germany), following the manufacturer’s instructions. For qPCR, a mixture containing 6.25 μL of TaqMan universal master mix II (Thermo Fisher, 4444965, Waltham, MA, USA), 0.5 μL of forward primer, 0.5 μL of reverse primer (Table 1) and 0.125 μL of the corresponding universal probe library (UPL) probe (Roche, 4683633001, Mannheim, Germany) (Table 1) for each reaction was prepared and dispatched in a 384-well plate. Then, 12.5 ng of cDNA in 2.5 μL was added to each well. The samples were run in Roche’s LightCycler 480, with an activation step of 10 min at 95 °C, an amplification step repeated 50 times with each step consisting of 15 s at 95 °C and 1 min at 60 °C and a last cooling step at 40 °C. The primers and UPL probes used are indicated in Table 1. qPCR data were analysed considering the cycle threshold values obtained from the run, and the relative expression was assessed using the 2^−ΔΔCT^ method [36].

### 4.8. Protein Isolation and Western Blot

The lysis solution was prepared following the cOmplete™ lysis-M solution (Roche, 11697498001, Mannheim, Germany) manufacturer’s instructions and supplemented with PhosSTOP (Sigma-Aldrich, 93106075, Taufkirchen, Germany). The cell pellet was resuspended in 50 µL of the lysis solution, incubated on ice for 30 min and centrifuged at 4 °C for 15 min at 13,000 rpm; the supernatants were stored at −80 °C. Protein quantification was performed using the DC™ Protein assay kit II (Bio-Rad Laboratories, 5000112, Hercules, CA, USA). Then, 20 µg of proteins were loaded in the Mini-PROTEAN TGX precast gels (Bio-Rad Laboratories, 456-9036, Hercules, CA, USA), transferred on the Trans-Blot^®^ transfer system membranes (Bio-Rad laboratories, 1704156, Hercules, CA, USA) that were inserted in 15-mL conical tubes containing 5 mL of 1× TBST and rolled on a roller mixer for 5 min. The membranes were blocked for 1 h with 5 mL of blocking solution (2.5 g of albumin, 50-mL 1× TBST) and subsequently incubated overnight at 4 °C with the blocking solution containing the primary antibody. The following day, the membranes were washed out with 1× TBST for 5 min and incubated for 1 h with the blocking solution containing the secondary antibody. The membranes were washed with 1× TBST three times for 5 min and developed using the WEST-ZOL^®^ plus detection system (iNtRON Biotechnology, 16024, Seongnam, Kyonggi-do, South Korea). The membranes were occasionally stripped, incubating them for 45 min in stripping buffer (20-mL SDS 10%, 12.5-mL Tris HCl, 67.5-mL water, 0.8-mL ß-mercaptoethanol and a pH of 6.8). The following primary antibodies were used: anti-IRAK2 (Cell Signalling, 4367S, Danvers, MA, USA, 1:1000), anti-IL6 (Biozol, LS-C 165212, Eching, Bayern, Germany, 1:1000), anti-cyclin D1 (Invitrogen, PA532373, Thermo Fisher, Waltham, MA, USA, 1:1000), anti-ERN1 (Proteintech, 27528-1, Rosemont, IL, USA, 1:1000), anti-CHOP (Sigma-Aldrich, SAB5700602, Taufkirchen, Germany, 1:1000), anti-BAK1 (Thermo Fisher, MA5-32111, Waltham, MA, USA, 1:1000), anti-ERK (Cell Signalling, 4696S, Danvers, MA, USA, 1:1000) and anti-P-ERK (Cell Signalling, 9101S, Danvers, MA, USA, 1:1000). The following secondary antibodies were used: polyclonal goat anti-rabbit (Dako Agilent, P044801, Santa Clara, CA, USA, 1:2000) and polyclonal rabbit anti-mouse (Dako Agilent, P0260, Santa Clara, CA, USA, 1:2000).

### 4.9. ER-Stress Induction

Control and shIRAK2 BCSC1 and BCSC3 were seeded at 1 × 10^5^ cells/well in a 6-well plate, and the medium was supplemented with 100-ng/mL doxycycline for 3 days. The cells were subsequently treated for 3 h with 5-µM thapsigargin in 2 mL of medium (Sigma-Aldrich, T9033, Taufkirchen, Germany). The cells were detached and centrifuged, and RNA was subsequently isolated from them.

### 4.10. Autophagy Evaluation

IRAK2 knockdown was induced with 100-ng/mL doxycycline in the control and shIRAK2 BCSC1 and BCSC3 for 4 days. Next, 1 × 10^4^ cells/well were seeded in a 96-well culture plate, treated overnight with 10-μM chloroquine (Sigma-Aldrich, C6628-25G, Taufkirchen, Germany) and subsequently fixed. LC3-II immunofluorescence staining was quantified using the ImageJ plugin automatic nuclei counter ITCN. Channel images were split, green images (LC3-II staining) were inverted and, using the plugin ITCN, the width (5 for BCSC1 and 6 for BCSC3), minimum distance (2.5 for BCSC1 and 3 for BCSC3) and threshold (1.5) to consider were defined. ‘Detect dark peaks’ was selected, and spots (representing LC3-II staining) were counted. Subsequently, nuclei in the blue images (DAPI staining) were manually counted. Autophagy was quantified by evaluating the ratio between the counted dark peaks over the number of nuclei per picture.

### 4.11. Apoptosis Assay

Control and shIRAK2 BCSC1 and BCSC3 were seeded at 1 × 10^4^ cells/well in 96-well plates in the presence of 100-ng/mL doxycycline for 5 days. Next, apoptosis was assessed using the apoptosis assay kit (Abcam, ab176749, Cambridge, UK) following the manufacturer’s instructions. Apoptosis was quantified considering the green fluorescence measured at Ex/Em = 490/525 nm over cellular confluence, both measured using the IncuCyte^®^ Live-Cell analysis system.

### 4.12. NF-κB Phosphorylation Assay

Control and shIRAK2 BCSC1 and BCSC3 were treated with 100-ng/mL doxycycline for 4 days and seeded at 3 × 10^4^ cells/well in 96-well culture plates. The EnzyFluo phosphorylation assay kit (BioAssay Systems, ABIN5691837, Hayward, CA, USA) was used to quantify the phosphorylated and total NF-κB, following the manufacturer’s instructions. NF-κB activation was defined by normalising the signal of the measured phosphorylated NF-κB to the NF-κB total protein content.

### 4.13. Orthotopic Breast Cancer Xenografts

Here, 1 × 10^5^ BCSC1 or 5 × 10^5^ MDA-MB-468 were mixed with one million irradiated Hs27 fibroblasts (ATCC, RRID:CVCL_0335, Manassas, VA, USA). Matrigel was added at a 1:1 ratio to the mixture for a total volume of 40 μL, and the mixtures were transplanted into the mammary fat pads of non-obese diabetic/severe combined immunodeficiency female mice. Two injections were performed for each mouse, one on each side. When the xenografts reached 3 mm in diameter, IRAK2 knockdown was induced by feeding the animals three times a week with gelatin containing 2-mg/mL doxycycline and 0.08-g/mL sugar. The animals were sacrificed when the xenografts reached a diameter close to 15 mm. Three mice received control cells and another three received IRAK2 knockdown cells.

### 4.14. Statistical Analysis

Data are expressed as the mean of the group plus the SEM, and statistical analysis was performed using the software GraphPad Prism 9 (GraphPad Software Inc., San Diego, CA, USA). Normality was assessed using the Shapiro–Wilk test. In the case of normal distribution, the unpaired Student’s t-test was performed, or else the non-parametric Mann–Whitney U test was used. The paired Student’s *t*-test was used to compare the growth of control and IRAK2 knockdown xenografts. Proliferation data were analysed using the Wilcoxon test. *p*-values less than 0.05 were considered as significant (*, *p* ≤ 0.05; **, *p* ≤ 0.01; ***, *p* ≤ 0.001; ****, *p* ≤ 0.0001).

## Figures and Tables

**Figure 1 ijms-24-02520-f001:**
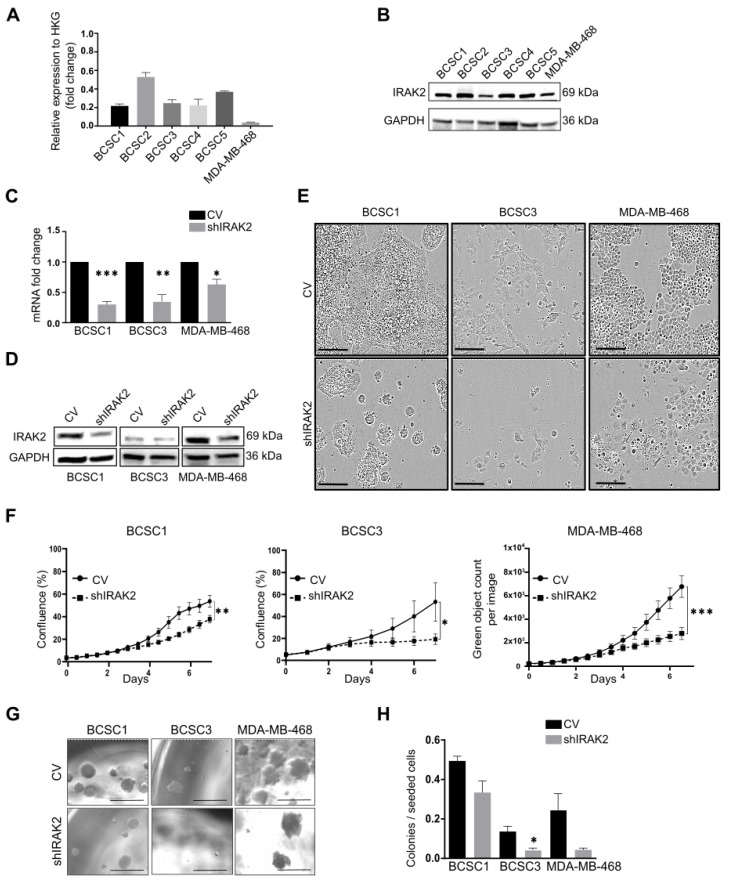
IRAK2 knockdown decreases the proliferation and sphere-forming capacity of BCSC1, BCSC3 and MDA-MB-468. (**A**) IRAK2 expression at the mRNA level in BCSCs and MDA-MB-468 assessed using quantitative real-time polymerase chain reaction (qPCR) analysis. IRAK2 relative expression was calculated using the 2^−ΔΔCT^ method and normalised to the expression of the housekeeping gene (HKG) hypoxanthine-guanine phosphoribosyl transferase (HPRT1). (**B**) IRAK2 expression in BCSCs and MDA-MB-468 at the protein level assessed by Western blot. (**C**) IRAK2 expression by qPCR in BCSC1, BCSC3 and MDA-MB-468 CV and shIRAK2 cells cultured for 5 days in the medium containing 100-ng/mL doxycycline. IRAK2 expression was normalised to HPRT1, and the relative expression was calculated using the 2^−ΔΔCT^ method and represented as the normalised quantity of mRNA in shIRAK2 cells relative to the normalised quantity of mRNA in control cells, setting the control as 1. (**D**) IRAK2 expression was assessed by Western blot in BCSC1, BCSC3 and MDA-MB-468 CV and shIRAK2 cells cultured for 5 days in a medium containing 100-ng/mL doxycycline. (**E**) Representative images of BCSC1, BCSC3 and MDA-MB-468 phenotypes in two-dimensional (2D) culture in control cells (CV) and IRAK2 knockdown cells (shIRAK2). Pictures were taken after 7 days of doxycycline-mediated knockdown induction. Scale bars represent 200 μm. (**F**,**G**) Reduced proliferation (**F**) and sphere-forming capacity (**G**) of cells characterised by IRAK2 knockdown compared with the control cells. Scale bars represent 500 μm. (**H**) Sphere-forming capacity quantification was defined as the ratio between the spheres counted in each well 14 days after the seeding, and the cells seeded per well. Proliferation data were analysed using the Wilcoxon test. Statistical significance for qPCR and sphere-forming capacity were performed using unpaired Student’s t-test. Data (n = 3) are presented as means ± standard error of the mean (SEM). *, *p* < 0.05; **, *p* < 0.01; ***, *p* < 0.001.

**Figure 2 ijms-24-02520-f002:**
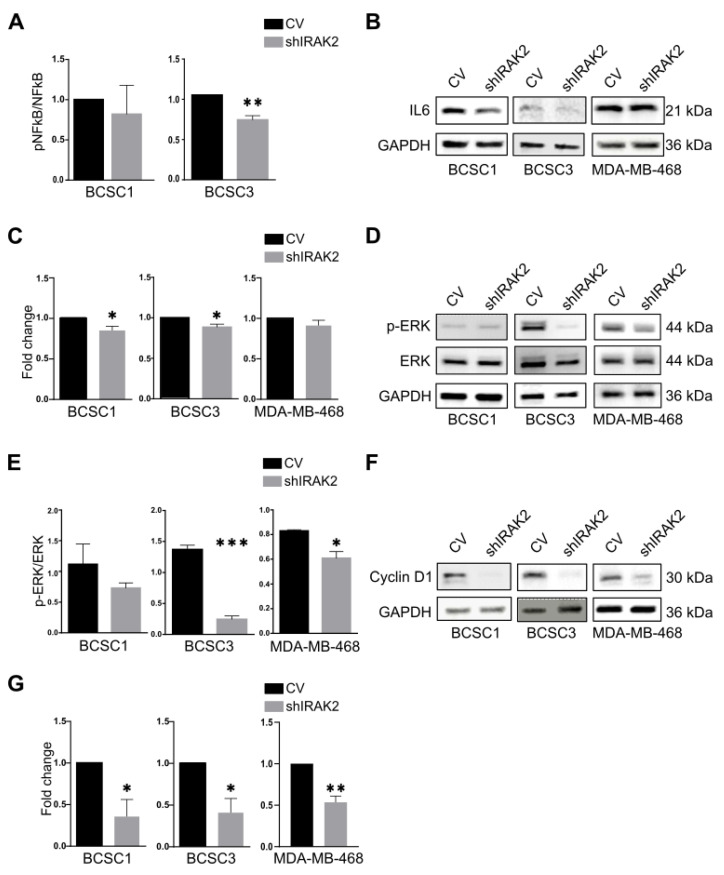
NF-κB and ERK phosphorylation and IL-6 and cyclin D1 expression are reduced by IRAK2 knockdown. (**A**) NF-κB phosphorylation in BCSC1 and BCSC3, comparing control (CV) and IRAK2 knockdown cells (shIRAK2). The NF-κB activity was assessed after culturing the cells for 5 days in the presence of doxycycline and using the EnzyFluo phosphorylation assay kit (BioAssay Systems, ABIN5691837). The NF-κB activity was defined by normalising the signal of phosphorylated NF-κB (pNF-κB) to the signal of the NF-κB total protein content. Data (n = 3) are presented as means ± SEM; statistical analysis was performed using unpaired Student’s t-test. (**B**–**G**) IL-6 expression (**B**) and its quantification (**C**), ERK phosphorylation (pERK) and total protein content (**D**) and its quantification (**E**), cyclin D1 expression (**F**) and its quantification (**G**) assessed using Western blot in BCSC1, BCSC3 and MDA-MB-468 CV and shIRAK2 cells cultured in doxycycline-containing medium for 5 days. Data are presented as means ± SEM; statistical analysis was performed using unpaired Student’s *t*-test. *, *p* < 0.05; **, *p* < 0.01; ***, *p* < 0.001.

**Figure 3 ijms-24-02520-f003:**
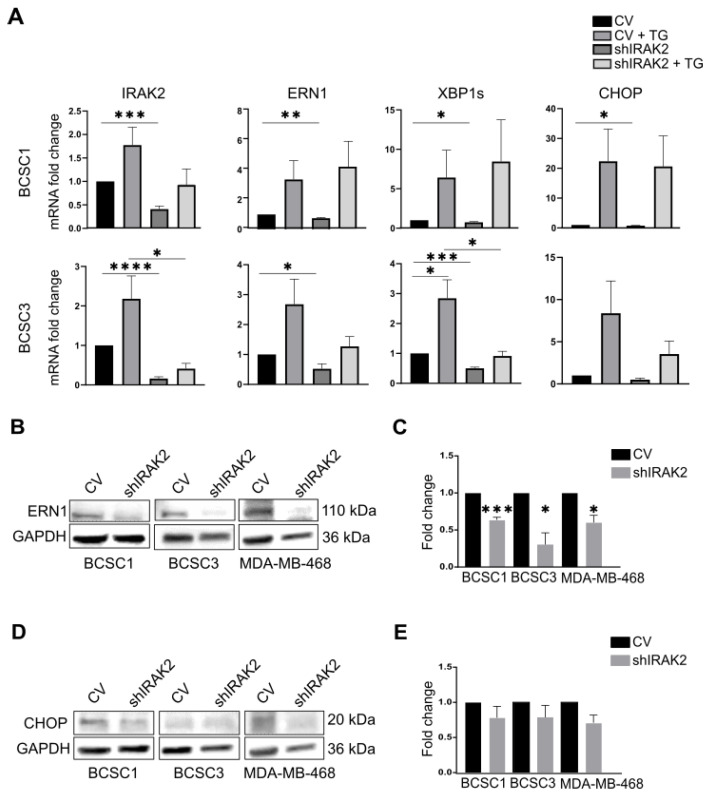
In the presence of ER-stress, IRAK2 is upregulated, and its downregulation affects the ERN1 pathway. (**A**) Expression of IRAK2, ERN1, XBP1s and CHOP at the mRNA level in control and IRAK2 knockdown BCSC1 and BCSC3. The transcription of the genes was evaluated in control cells (CV), control cells treated with thapsigargin (CV + TG), IRAK2 knockdown cells (shIRAK2) and IRAK2 knockdown cells treated with thapsigargin (shIRAK2 + TG). All cells (thapsigargin-treated and untreated, control and shIRAK2 cells) were cultured in the presence of doxycycline. Cells were harvested, and RNA was isolated from their pellet. The gene of interest (IRAK2, ERN1, XBP1s and CHOP) expression was normalised to the HKG β-actin. The relative expression was calculated using the 2^−ΔΔCT^ method and represented as the normalised quantity of mRNA in shIRAK2 cells, CV + TG cells and shIRAK2 + TG cells relative to the normalised quantity of mRNA in CV cells, which was set as 1. (**B**–**E**) The expression of ERN1 (**B**) and CHOP (**D**) at the protein level and their relative quantification ((**C**) for ERN1 and (**E**) for CHOP) in the CV and shIRAK2 BCSC1, BCSC3 and MDA-MB-468. Cells were cultured for 5 days in a medium containing 100-ng/mL doxycycline, harvested, and the proteins were subsequently isolated from the cell pellet. The expression of the gene of interest (ERN1 or CHOP) was normalised to the expression of glyceraldehyde-3-phosphate dehydrogenase (GAPDH). Subsequently, the expression evaluated in shIRAK2 cells was compared with that calculated in CV cells, which was set as 1. Data (n = 3) are presented as means ± SEM using unpaired Student’s *t*-test. *, *p* < 0.05; **, *p* < 0.01; ***, *p* < 0.001; ****, *p* < 0.0001.

**Figure 4 ijms-24-02520-f004:**
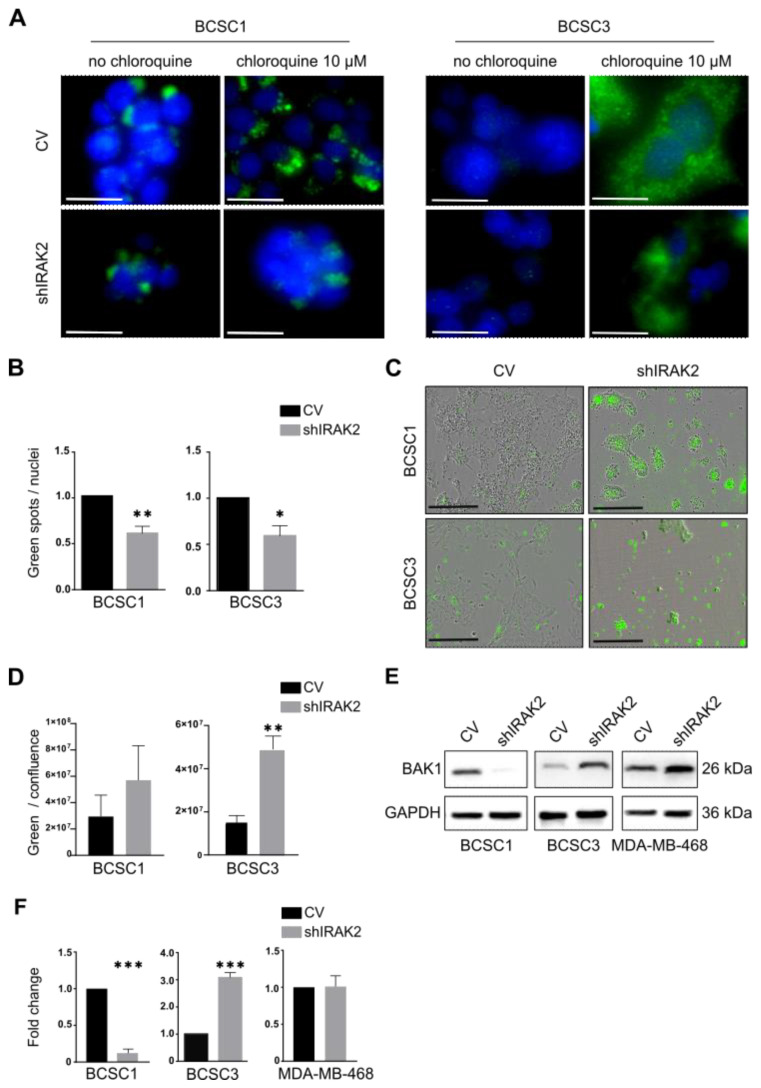
IRAK2 knockdown decreases autophagy in BCSC1 and BCSC3 and induces apoptosis in BCSC3. (**A**) Immunofluorescence staining (green fluorescence) of LC3-II in control (CV) and IRAK2-induced knockdown (shIRAK2) BCSC1 and BCSC3 cells treated with chloroquine (chloroquine 10 µM) or not (no chloroquine). 4′,6-diamidino-2-phenylindole allowed visualisation of the nuclei in blue. Cells were cultured for 5 days in a medium supplemented with 100-ng/mL doxycycline, treated overnight with 10-µM chloroquine and fixed, and LC3-II immunofluorescence staining was performed. Scale bars represent 25 µm. (**B**) Autophagy was quantified considering the ratio between the counted green spots representing the LC3-II over the number of nuclei counted per picture. (**C**) Representative pictures of phosphatidylserine (PS) externalisation (green fluorescence) in CV and shIRAK2 BCSC1 and BCSC3. Cells were cultured for 5 days with 100-ng/mL doxycycline and subsequently treated to visualise the PS externalisation. Scale bars represent 400 µM. (**D**) Apoptosis quantification was expressed as the ratio of green fluorescence over cell confluence. (**E**,**F**) BAK1 expression at the protein level in CV and shIRAK2 BCSC1, BCSC3 and MDA-MB-468 (**E**) and relative quantification (**F**). Cells were cultured for 5 days with 100-ng/mL doxycycline and harvested, and proteins were isolated. BAK1 expression was normalised to the GAPDH expression. Then, the expression evaluated in shIRAK2 cells was compared with that calculated in CV cells, which was set as 1. All the data in these graphs (n = 3) are presented as means ± SEM using unpaired Student’s *t*-test. *, *p* ≤ 0.05; **, *p* ≤ 0.01; ***, *p* < 0.001.

**Figure 5 ijms-24-02520-f005:**
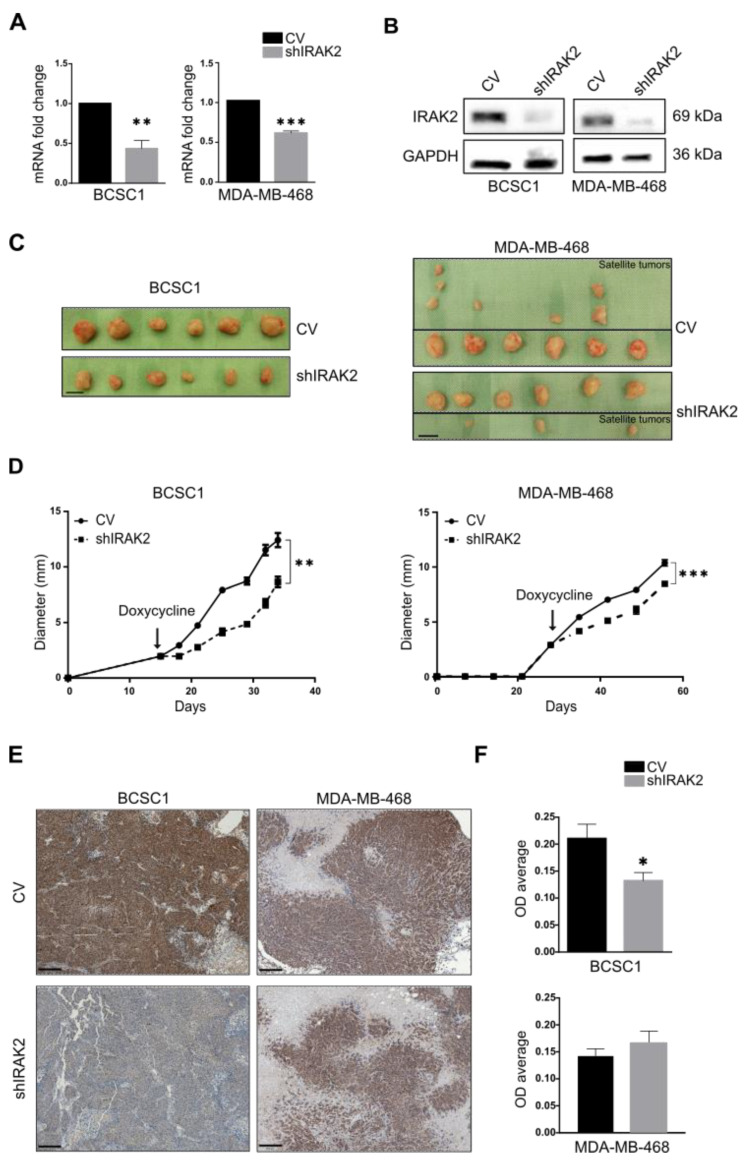
The ability of BCSC1 and MDA-MB-468 to form tumours is decreased by IRAK2 downregulation. (**A**) IRAK2 expression was assessed in xenografts derived from control (CV) and IRAK2 knockdown (shIRAK2) BCSC1 or MDA-MB-468 cells at the mRNA level using qPCR. IRAK2 expression was normalised to β-actin; the relative expression was calculated using the 2^−ΔΔCT^ method and represented as the normalised quantity of mRNA in shIRAK2 xenografts relative to the normalised quantity of mRNA in control xenografts, setting the control as 1. Data (n = 3) are presented as means ± SEM; statistical analysis was performed using unpaired Student’s *t*-test. (**B**) IRAK2 expression in CV and shIRAK2 cell-derived xenografts at the protein level evaluated using Western blot. (**C**) BCSC1 and MDA-MB-468 CV and shIRAK2 xenografts. Regarding MDA-MB-468, the upper panels show control xenografts with the corresponding satellite tumours, whereas the lower ones show shIRAK2 xenografts with the related satellite tumours. Scale bars represent 1 cm. (**D**) Xenograft diameter over time. The arrows indicate the doxycycline feeding starting time. Data (n = 6) are presented as means ± SEM; statistical significance was evaluated using paired Student’s *t*-test. (**E**) Immunohistochemistry representing cyclin D1 expression in CV and shIRAK2 xenografts derived from BCSC1 and MDA-MB-468. Scale bars represent 200 µm. (**F**) Relative quantification of the immunohistochemistry staining expressed as optical density (OD) average. Five xenografts per condition were considered (n = 5). One slide for xenograft was stained. Four to five pictures were taken for each slide and quantified. First, the OD average of the pictures taken from the same slide was evaluated. Subsequently, OD averages for control xenografts and IRAK2 knockdown xenografts were calculated and compared. Statistical significance was evaluated using unpaired Student’s *t*-test. *, *p* < 0.05; **, *p* < 0.01; ***, *p* < 0.001.

**Table 1 ijms-24-02520-t001:** Primers and universal probe library (UPL) probes used in qPCR experiments.

Gene	Forward (5′-3′)	Reverse (5′-3′)	UPL
ACTB	CCAACCGCGAGAAGATGA	CCAGAGGCGTACAGGGATAG	#64
CHOP	AAGCAGCGCATGAAGGAG	GCCGTTCATTCTCTTCAGCTA	#2
ERN1	CTGCCCATCAACCTCTCTTC	AGCTCTCGGGTTTTGGTGT	#9
HPRT	TGACCTTGATTTATTTTGCATACC	CGAGCAAGACGTTCAGTCCT	#73
IRAK2	ATTCTTCCAGGCAGAGTTGC	GCCCAGCACAGGTAAGACAT	#87
XBP1 spliced form	AGTTAAGACAGCGCTTGGGG	TGCACCTGCTGCGGACTCAG	#37

## Data Availability

Data that support the findings of this study are available from the corresponding author upon request.

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
