# Peer review of "IRAK2 Downregulation in Triple-Negative Breast Cancer Cells Decreases Cellular Growth In Vitro and Delays Tumour Progression in Murine Models"

_ijms, 2023, doi:10.3390/ijms24032520_

Round 1

Reviewer 1 Report

The manuscript by Ferraro et.al, identifies and summarizes the results which were obtained after IRAK2 was downregulated in TNBC.  IRAK2 is a component of the interleukin 1 receptor/ toll like receptor signaling cascade. The authors downregulated IRAK2 in breast cancer stem cells  as well as MDA-MB-468 cells using lentiviral mediated shRNA targeting. The authors wanted to assess if this downregulation will lead to any change in the cellular phenotype, proliferation, sphere formation capacity. The authors also assessed tumor growth.  They found that IRAK2 downregulation results in decreased proliferation and sphere formation.  Similarly, IRAK2 downregulation mitigated ERN1 signaling and autophagy pathways. The downregulation of IRAK2 also decreased NF-kB and ERK phosphorylation as well as IL-6 and cyclin D1 expression in BCSCs and MDA-MB-468 cells. It also resulted in a delay in the growth of xenografts. Overall, a very well written and detailed manuscript. The manuscript organization is easy to follow and well structured. The language is simple and easy to understand. All the experiments done are of good quality and results support their hypothesis.

About the themethodology, the authors can in future perform the RNA seq or RPPA validation to get the complete picture of all the pathways involved . Also, they can use MCF10a as a control which is a non malignant breast cancer cell line.

Author Response

We thank Reviewer 1 for taking the time to read our work, positive evaluation and very helpful insights and comments. We are grateful for the interesting suggestions, that will be considered for future studies, which we will perform with additional funding.

Reviewer 2 Report

The impact of IRAK2 (interleukin-1 receptor-associated kinase 2) downregulation on breast cancer stem cells derived from TNBCs has been investigated by Fancesca Ferraro et al. in this manuscript. The results were then compared to the established TNBC cell line MDA-MB-468. The study shows that sphere-forming ability and cellular proliferation are both impacted by IRAK2 knockdown. This was confirmed by decreased tumor growth in the xenograft model with IRAK2 downregulation.

This study builds on a previous study in which the authors established BCSCs and used kinase assays to identify specific kinases that were highly enriched in MDA-MB-468 and BCSCs.

I have a few questions,

1.      Since this study is connected to earlier work, a brief explanation of how BCSCs are generated using TNBC and a kinase assay will be helpful for follow-up.

2.      Given their propensity for adaptation and change, the established BCSCs should be once more assessed for their stem cell characteristics. Experiments such as the ALDH assay and CD44/CD24 flow cytometry should be performed to confirm the stem cell population.

3.      How does the author interpret the effect of IRAK2 knockdown on cellular phenotype when it shows a differential effect? It is challenging to comprehend the significance of Figure 1C.

4.      Why did the author choose BCSC1, BCSC3, and MDA-MB-468? BCSC1 and 3 have lower IRAK2 expression than other BCSCs, and MDA-MB-468 has the lowest expression. BCSCs or TNBC cell lines with higher IRAK2 expression should have been included for a better understanding.

5.      I was unable to locate Figure A1A and related keratin expression results, as mentioned in lines 99-104.

6.      The Western-blotting images in figures 3B and D are unclear, and it is challenging to understand any particular effect in figure 3D.

7.      The immunofluorescence image in figure 4A is pixelated and should be improved.

8.      The author mentioned that IRAK2 is upregulated during ER stress. Cellular proliferation and growth were inhibited by IRAK2 knockdown, indicating cellular stress. How does the author explain that in IRAK2 knockdown cells, ER stress caused by thapsigargin increases the expression of IRAK2?

9.      The manuscript should be copy-edited because there are several grammatical errors, and sentences should be rephrased for clarity.

Round 2

Reviewer 2 Report

Since the majority of the earlier concerns have been addressed, I have nothing further to add.